# Synergistic Design of Anatase–Rutile TiO$_2$ Nanostructured Heterophase Junctions toward Efficient Photoelectrochemical Water Oxidation

**Sangwoo Lee** [1],[†] , **A. Young Cho** [1],[†], **You Seung Rim** [2], **Jun-Young Park** [1] **and Taekjib Choi** [1],*

[1] Hybrid Materials Research Center and Department of Nanotechnology and Advanced Materials Engineering, Sejong University, Seoul 05006, Korea; ls3210@naver.com (S.L.); dkdud5372@naver.com (A.Y.C.); jyoung@sejong.ac.kr (J.-Y.P.)

[2] Department of Intelligent Mechatronics Engineering, Seoul 05006, Korea; youseung@sejong.ac.kr

* Correspondence: tjchoi@sejong.ac.kr

† These authors contributed equally to this work.

**Abstract:** Synergistically designing porous nanostructures and appropriate band alignment for TiO$_2$ heterophase junctions is key to efficient charge transfer, which is crucial in enhancing photoelectrochemical (PEC) water splitting for hydrogen production. Here, we investigate the efficiency of PEC water oxidation in anatase–rutile TiO$_2$ nanostructured heterophase junctions that present the type-II band alignment. We specifically prove the importance of a phase alignment in heterophase junction for effective charge separation. The TiO$_2$ heterophase junctions were prepared by transferring TiO$_2$ nanotube (TNT) arrays onto FTO substrate with the help of a TiO$_2$ nanoparticle (TNP) glue layer. The PEC characterization reveals that the rutile (R)-TNT/anatase (A)-TNP heterophase junction has a higher photocurrent density than those of A-TNT/R-TNP junction and anatase or rutile single phase, corresponding to twofold enhanced efficiency. This type-II band alignment of R-TNT/A-TNP for water oxidation, in which photogenerated electrons (holes) will flow from rutile (anatase) to anatase (rutile), enables to facilitate efficient electron-hole separation as well as lower the effective bandgap of heterophase junctions. This work provides insight into the functional role of heterophase junction for boosting the PEC performances of TiO$_2$ nanostructures.

**Keywords:** photoelectrochemical; solar-water splitting; TiO$_2$ heterophase junctions; TiO$_2$ nanostructure

## 1. Introduction

Semiconductor nanostructure-based photoelectrochemical (PEC) or photocatalytic (PC) water splitting is an emerging technology in energy conversion, benefiting from its possible high efficiency. The diverse one-dimensional (1D) nanostructures of a semiconductor have been extensively explored for heterogeneous photocatalysis and heterostructure-based PEC photoelectrodes [1–3]. Among the 1D nanostructured semiconductors, the TiO$_2$ nanotube (TNT) has attracted much interest as a promising photocatalytic material for energy-conversion applications owing to its superior physicochemical properties, such as excellent chemical stability and remarkable catalytic properties [4,5]. Such unique properties, as compared to nanoparticles or bulk materials, can offer larger specific surface area, faster electron transport, and enhanced light absorption/scattering, which are beneficial for effective PEC water splitting. However, the PEC or PC performance of single-phase TiO$_2$ nanostructures remains limited due to the large bandgap harnessing of the UV region of solar light and rapid recombination of photogenerated charge carriers [6,7]. In general, TiO$_2$ can exist in three distinct polymorphs: anatase, rutile, and brookite [8,9]. Rutile is the thermodynamically stable form in a tetragonal structure, while metastable -anatase and -brookite, having an orthorhombic structure, can

be irreversibly transformed to rutile at elevated temperatures. Among them, anatase $TiO_2$ is the most active photocatalyst because of its appropriate electronic band structure as well as slow recombination of the photogenerated electron-hole pairs, although anatase $TiO_2$ has a larger bandgap (~3.2 eV) than does the rutile $TiO_2$ (~3.0 eV); it could be primarily active for UV light [10,11]. While rutile $TiO_2$ usually exhibits a less-efficient PC performance due to its higher recombination rate of electron-hole pairs [12], in order to overcome the shortcomings of single-phase $TiO_2$ nanostructures, it is highly desirable to employ many strategies, including designing of heterojunctions [13], engineering of surface morphology [14], and heteroatom-doping [15,16].

Numerous approaches have been developed for extending the photoresponse range, promoting the photogenerated charge transfer/separation or inhibiting the photoexcited charge recombination, and increasing the surface reaction sites [17]. Among these approaches, fabrication of $TiO_2$ heterostructures by coupling with other semiconductors has attracted great attention due to their excellent effectiveness in improving the separation of photogenerated charge and enhancing visible-light absorption [18–20]. Representatively, construction of the p-n heterojunction can provide a strong internal electric field due to the large Fermi-level difference in two semiconductors, which is a common approach toward improving the efficiency of charge carrier separation. Various p-n heterojunction photocatalysts, such as $NiO/TiO_2$ [21], $Cu_2O/TiO_2$ [22], $SnO/TiO_2$ [23], and $MoS_2/TiO_2$ [24], have been developed. Wang et al. reported that well-defined $NiO/TiO_2$ hollow hybrid shells with large surface area show remarkably improved PC $H_2$ production with 2.1% apparent quantum efficiency under visible light irradiation by coupling n-type $TiO_2$ with p-type NiO. Meanwhile, formation of the type-II heterojunctions is an effective strategy for enhancing the charge separation efficiency, resulting from the built-in electric field due to band bending at the interface of two semiconductors. Cheng et al. demonstrated that the improved PEC performance of planar $BiVO_4/TiO_2$ heterojunction is caused by the formation of an energy-matched type-II junction [25]. The authors suggested that the recombination can be suppressed in the $BiVO_4$ layer due to the improvement of charge separation via formation of the heterojunction.

Thus, designing the junction is critical to boosting charge transfer/separation for enhanced PC activity. Notably, the heterophase junction, consisting of two different polymorphs of the same semiconductor, exhibits the type-II junction with staggered alignment of band edges at the interface. As compared with the heterojunction, heterophase junction is beneficial for photogenerated charge transfer at the interface, as the preparation of heterophase junctions is facile and controllable. The various heterophase $TiO_2$ materials, such as biphasic or even three mixed phases of anatase, rutile, and brookite $TiO_2$, have been explored for energy conversion applications, which present higher PC or PEC activities than that of single-phase $TiO_2$ [26–28]. For comparison, a brief summary of previously reported PEC performances in $TiO_2$ heterophase based photoanodes is provided in Table S1. Several studies have been performed to determine the role of heterophase junctions in $TiO_2$ nanoparticles, nanofiber, and nanorods. However, their results are still limited to present the functional role of a heterophase junction in charge separation.

The higher PEC performance mainly results from significantly lowering the effective bandgap of composite materials and facilitating efficient charge carrier separation due to a type-II band alignment of phase junctions. For anatase–rutile $TiO_2$ heterophase photocatalysts, which are composed of anatase–rutile $TiO_2$ composite nanoparticles [29,30], the different bandgaps and aligned band-edge positions of anatase and rutile $TiO_2$ derive that photogenerated electrons will flow from rutile to anatase. This appropriate phase alignment leads to desirable charge separation for the enhanced PC activity of anatase–rutile composites over pure anatase or rutile $TiO_2$.

Nevertheless, PC activity of anatase–rutile $TiO_2$ composite nanoparticles is still impeded by the small specific surface area and limited surface active sites. In this respect, developing $TiO_2$-based heterophase junctions that have the larger specific surface area provided by nanostructures with high porosity and long aspect ratio and an efficient charge transfer/separation offered by a favorable band alignment would provide a feasible route toward highly efficient PEC water oxidation. In this work,

by utilizing $TiO_2$ nanostructures, such as TNT and TNP, we fabricated TNT/TNP heterophase junctions with different phase sequence, i.e., A-TNT/R-TNP and R-TNT/A-TNP junctions. We investigated the relationship among surface morphology of nanostructure, optical, and band alignment properties with respect to probe proper phase sequence in heterojunctions, in which photogenerated electrons would flow from rutile to anatase. The anatase–rutile heterophase junctions, considered to be type-II band structure, can lower the effective bandgap (corresponding to ~2.9 eV). Especially, the R-TNT/A-TNP junction exhibited superior PEC performance, which is one order than that of R-TNP and two times higher than A-TNP. Such enhanced PEC or PC properties are responsible for the efficient charge separation due to optimal band alignment and the enhanced surface catalytic activity due to 1D nanostructured TNT with a hierarchical porous structure.

## 2. Experimental Procedures

### 2.1. Preparation of the $TiO_2$ Heterophase Junction Photoelectrode

The highly ordered and hierarchical $TiO_2$ nanotube (TNT) arrays featuring nanopores or nanotubular structures with high aspect ratios were prepared by the two-step anodization process of Ti foils (99.9% pure) in ethylene glycol (99.5%, Samchun) with 0.3 wt% $NH_4F$ (ammonium fluoride ACS reagent, 98.0%, Sigma Aldrich) and $H_2O$ at concentration of 2%. For the first step of anodization, Ti foil was anodized for 12 h at 60 V and 15 °C; then, the as-anodized sample were rinsed in D.I. water and ultrasonically dissolved in a mixed solution (hydrogen peroxide: D.I. water = 1:1) for 15 min. The second anodization step was performed for 1 h under the same condition. The amorphous TNT/Ti were annealed to yield the anatase phase at 500 °C for 1 h in air. After annealing, the as-annealed sample was anodized for 15 min; then, we performed a detaching process in aqueous $H_2O_2$. As a phase transformation method, the rapid thermal annealing treatment was used to tune anatase and rutile phase configurations of the detached TNTs at the elevated temperatures up to 900 °C for 30 s. The 5 μm thick crystalline TNT films were attached on FTO (fluorine-doped tin oxide) glass using $TiO_2$ nanoparticles paste (TNP) with 5 μm thickness applied by doctor-blade method and followed by annealing at 500 °C for 1 h in air. Finally, we obtained the various heterophase nanostructured photoelectrodes, such as anatase-TNT/rutile-TNP, rutile-TNT/anatase-TNP, and rutile-TNP/anatase-TNP structures, deposited on FTO glass.

### 2.2. Structural and Optical Characterizations of the $TiO_2$ Nanostrucrtures

The structural and morphological characterizations of the $TiO_2$ nanostructures were carried out using a field emission scanning electron microscopy (FE-SEM, Hitachi S-4700, Tokyo, Japan). Detection of crystalline phases was performed via high-resolution X-ray diffraction (XRD, Rigaku D/Max-2500, Tokyo, Japan) with monochromatic Cu Kα radiation (λ = 1.5418 Å). The light absorption spectra of the prepared samples were collected with an ultraviolet-Vis spectrophotometer (Cary 5000, Agilent, Santa Clara, CA, USA). The crystal structure of the corresponding samples was characterized with Raman spectroscopy (Renishaw inVia, New Mills, UK, He–Ne laser wavelength of 633 nm) equipped with an optical microscope. The chemical bonding states of the $TiO_2$ nanostructures were analyzed by X-ray photoelectron spectroscopy (XPS, PHI 5000 VersaProbe, Physical Electronics Inc., Chanhassen, MN, USA) with an Al Kα X-ray source. For the binding energy correction, the C 1s (284.8 eV) signal was used as a charge correction reference.

### 2.3. Photoelectrochemical Testing

The photoelectrochemical (PEC) performances were evaluated by using a potentiostat (Ivium vertex, Netherlands) with a three-electrode configuration: a Pt counter-electrode, a saturated Ag/AgCl (3 M NaCl) reference electrode, and the $TiO_2$ heterophase or $TiO_2$ single-phase working electrode were used in a 1 M KOH electrolyte at room temperature. The current density-potential (*J-V*) curve of the scan rate was 50 mV/s. The light source was a 150 W Xenon lamp, which was used to simulate the 1

sun irradiation condition (AM 1.5 G, 100 mW cm$^{-2}$) by using a solar simulator (Oriel 94021A, Irvine, CA, USA).

## 3. Results and Discussion

As a photoanode for the photoelectrochemical cell, we used TiO$_2$ nanostructures with single phase and heterophase because of their excellent chemical stability and strong surface catalytic activity, in which photogenerated holes (electrons) will participate in the water oxidation reaction (proton reduction reaction) at the anode (cathode)-electrolyte interface. Two different such reactions take into consideration the charge separation/transport within the photoanode for PEC water splitting. In particular, as illustrated in Figure 1, TiO$_2$ nanostructured heterophase junctions were fabricated by process steps comprising of (1) via two-step anodization, TNTs were prepared on Ti foil-forming hierarchical 1D vertical nanostructures (see SEM images of TNT for top surface and bottom surface); (2) detachment of the TNT film from the Ti foil and annealing process for crystallization and phase transformation; (3) TNT/TNP bilayer junctions consisting of an internal TNP and external TNT were successfully fabricated by transferring and attachment of TNT layer onto FTO glass coated with a TNP paste (see the cross-sectional SEM image of TNT/TNP junction). We also confirmed that the TNT layer has a clear periodic nanoring structure on the top and highly aligned vertical nanotubes underneath with an average diameter of about 100 nm and wall thickness of about 20 nm. The hierarchical TNT led to improved PEC activities because of the high specific surface areas, the high light harvesting efficiency due to larger light scattering, and the high electron transport provided through 1D nanostructures.

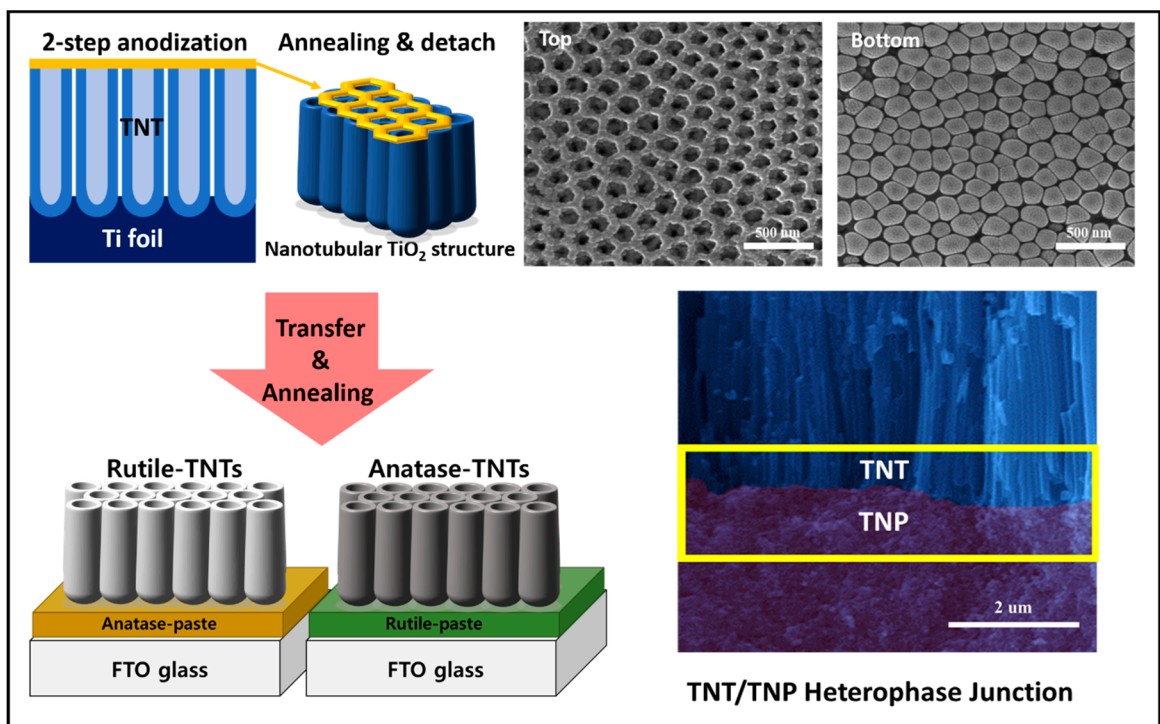

**Figure 1.** Schematic illustration of fabrication process for the hierarchical TiO$_2$ heterophase junction.

Raman scattering spectroscopy represents a powerful method for studying the structural-compositional properties of materials [31–34]. The TiO$_2$ polymorphs exhibit distinct Raman spectra, so that Raman scattering spectroscopy enables precise determination of the TiO$_2$ crystal phases. The Raman spectra of pure A-TNP, R-TNP, and A-R heterophase of TNT/TNP are presented in Figure 2a. The frequency and symmetry assignment of the phonon modes are followed as: the E$_g$ peak is mainly caused by symmetric stretching vibration of O−Ti−O in TiO$_2$; the B$_{1g}$ peak is caused by symmetric bending vibration of O−Ti−O; and the A$_{1g}$ peak is caused by antisymmetric bending

vibration of O−Ti−O. It is well known that the anatase phase has six Raman active modes [32]: a peak with strong signal appears at $144 \pm 1$ cm$^{-1}$ (E$_g$) followed by low intensity peaks located at $196 \pm 1$ cm$^{-1}$ (E$_g$), $395 \pm 1$ cm$^{-1}$ (B$_{1g}$), $513 \pm 2$ cm$^{-1}$ (B$_{1g}$), $518 \pm 2$ cm$^{-1}$ (A$_{1g}$), and $639 \pm 2$ cm$^{-1}$ (E$_g$). The bands located at 513 and 518 cm$^{-1}$ overlap, and our equipment shows only one band at 515 cm$^{-1}$. Meanwhile, the rutile phase has four active modes appearing at $143 \pm 1$ cm$^{-1}$ (B$_{1g}$), $235 \pm 5$ cm$^{-1}$ (combination of phonon modes), $448 \pm 2$ cm$^{-1}$ (E$_g$), and $609 \pm 2$ cm$^{-1}$ (A$_{1g}$) [31]. The spectra were shortened and had their background removed by a straight line from 300 to 700 cm$^{-1}$, which proved to be the most appropriate for evaluating the rutile and anatase phases present in the TiO$_2$ nanostructures. Accordingly, Raman spectra of A-TNP are located at 396, 515, and 637 cm$^{-1}$. Related to R-TNP, the phonon modes appear at 447 and 610 cm$^{-1}$. The A-R heterophase junctions show the Raman spectra of anatase–rutile mixtures.

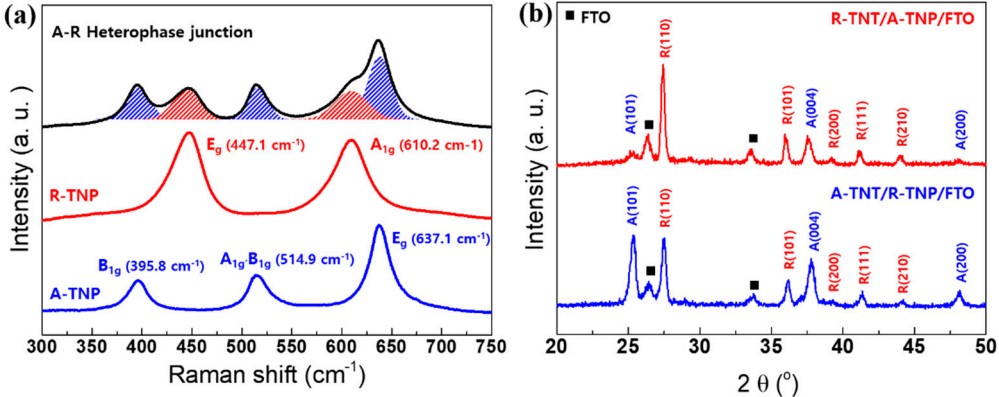

**Figure 2.** Structural characterizations of single-phase and heterophase TiO$_2$ nanostructured photoanodes: (**a**) Raman spectra of A-TNP, R-TNP, and R-TNT/A-TNP samples. (**b**) XRD patterns of two different heterophase junctions with different anatase–rutile phase sequences.

On the other hand, we studied the effect of annealing temperature on the microstructure for TNTs. The annealing temperature has a close relationship with crystallinity and crystal phase of the TNTs. To investigate crystalline structure evolution of TNTs with various annealing temperatures, we performed high-resolution x-ray diffraction for the annealed samples. The XRD patterns of the TNTs are shown in Figure S1. Characteristic peaks of anatase and rutile are indicated. For the TNTs annealed at 500 °C, the XRD pattern presents characteristic peaks of anatase crystal planes, (101), (004), and (200). However, we observed the main diffraction peak of the (101) plane at $2\theta = 25.4°$, indicating that TNTs have a highly crystalline anatase phase. It further suggests that, during annealing, crystal growth via interface nucleation at the nanotube bottoms adopt crystallographically specific orientations, e.g., plane (101). The metastable-anatase phase transforms irreversibly into rutile at the elevated temperatures. After higher-temperature annealing by rapid thermal treatment, TNTs tended to transform into the rutile phase. The diffraction peaks of anatase disappear at the higher temperatures, and the diffraction patterns of rutile phase become predominant. As a result, we observed the rutile peaks from the crystallographic planes (110), (101), and (111). At annealing temperatures higher than 800 °C, only rutile peaks appear. The XRD profiles of A-R heterophase junctions are presented in Figure 2b. We compare the XRD patterns of two different heterophase junctions with different anatase–rutile configuration. Both rutile–anatase (R-TNT/A-TNP) and anatase–rutile (A-TNT/R-TNP) heterophase junctions show the mixture of the anatase main diffraction peak of the (101) and (004) planes and the rutile main diffraction peak of the (110), (101), and (111) planes. We also investigated the crystallinity and d-spacing of anatase- and rutile-TiO$_2$ nanostructures in the heterophase junctions. The average crystallite size was estimated from the line broadening of x-ray diffraction peaks using the Sherrer formula as expressed by $D = k\lambda/(\beta\cos\theta)$, where $D$ is the crystallite size, $k$ is the Sherrer constant (0.9), $\lambda$ is the wavelength of the x-ray radiation (0.15418 nm for Cu K$\alpha$), and $\beta$ is the full width at half-maximum (FWHM) in x-ray diffraction reflections of the (101) and (110) peaks of anatase and rutile

TiO$_2$, respectively. Accordingly, the crystallites of TiO$_2$ nanostructures are summarized in Table S2. It is obvious that the crystallinities of R-TNT (26.444 nm) and A-TNT (17.645 nm) as an external layer are better than those of the R-TNP (21.900 nm) and A-TNP (11.348 nm) layer. The d-spacing of anatase (101) plane and rutile (110) plane are similar irrespective of TNP and TNT. Therefore, these Raman and XRD results confirm the successful fabrication of the A-R heterophase junctions.

X-ray photoelectron spectroscopy (XPS) was used to estimate the chemical bonding structures of the TiO$_2$ nanostructures. The full survey spectra of the samples reveal signals of C, Ti, and O elements, as shown in Figure S2. Figure S3a presents the O 1s XPS spectra of the A-TNT (R-TNT) samples, which were deconvoluted to two peaks at 530.77 (530.67) eV and 531.54 (531.17) eV, corresponding to the oxygen atoms O$^{2-}$ in the lattice and the absorbed –OH group, respectively. As can be seen in Figure S3b, the Ti 2p$_{3/2}$ and Ti 2p$_{1/2}$ peaks were observed at 459.49 eV (459.22 eV) and 465.20 eV (464.65 eV) for the A-TNT (R-TNT), which are assigned to Ti$^{4+}$. Figure S4 exhibits the elemental depth profiles of O1s and Ti2p for heterophase junctions performed by Ar$^+$ ion sputtering. The stoichiometry of the samples determined from the basis of the O/Ti intensity ratio are TiO$_{2.16}$ for R-TNT/A-TNP and TiO$_{2.05}$ for A-TNT/R-TNP, respectively.

The light-absorption properties of the R-TNT/A-TNP heterophase junction and single-phase A-TNP samples were examined by a UV-visible spectrophotometer. Figure 3a shows the comparative UV-visible absorption spectra of R-TNT/A-TNP and A-TNP. The absorption edge of A-TNP appears at around 385 nm. For the R-TNT/A-TNP heterophase junction, the absorption edge extends to 420 nm. Compared with A-TNP, the absorption edge of R-TNT/A-TNP exhibits an obvious redshift due to the formed anatase/rutile heterophase junction, which reveals the enhanced light-harvesting capability of the heterophase junction. The bandgap determines the optical absorption ability of a photoelectrode. Moreover, the appropriate band alignment of photoelectrodes are critical to achieve the high-performance PEC water splitting. We thus make use of Tauc plots to determine the optical bandgaps of single-phase A- and R-TNPs and A-R heterophase junctions. The relationship between the absorption coefficient and the incident photon energy is a Tauc plot relation via the following equation: $(\alpha h\nu)^{1/n} = A(h\nu - E_g)$, where $\alpha$ is the absorption coefficient of the material, $h$ is the Planck's constant, $\nu$ is the frequency, $E_g$ is the bandgap, $A$ is the proportionality constant, and $n$ is the value denoting the nature of the transition. The bandgap values ($E_g$) of A-TNP and R-TNP are estimated to be about 3.20 and 3.00 eV, respectively, while the light absorption capability of the heterophase junctions was confirmed as evident from the estimated bandgap of around 2.95 eV, irrespective of phase sequences (Figure 3b and Figure S5). Thus, when anatase and rutile phases are combined, the typical type-II band structure could be established at the phase junction interface. Thus, type-II junctions with a proper band alignment prevents the radiation recombination of photoexcited carriers and facilitates charge separation (Figure 4). The charge separation mechanism is supported by recent studies on the band alignment of anatase and rutile.

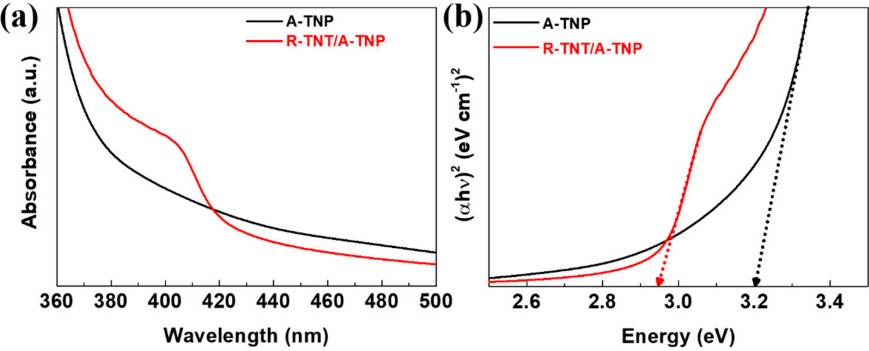

**Figure 3.** Light absorption capability and bandgap estimation for A-TNP and R-TNT/A-TNP: (**a**) UV-visible absorption spectra; (**b**) Tauc plots for A-TNP and R-TNT/A-TNP.

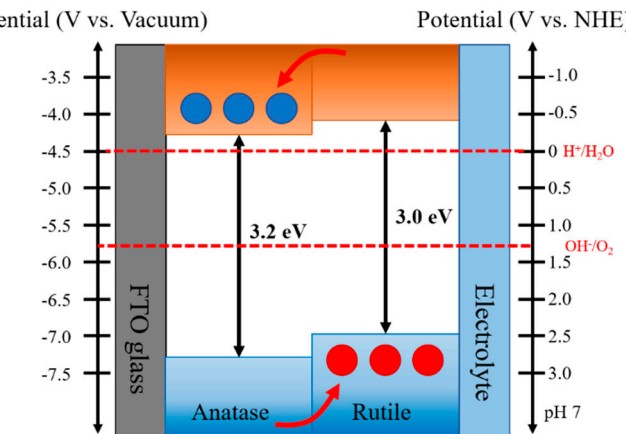

**Figure 4.** Schematic of the type-II band alignment staggered with the conduction band and valence band energy levels of rutile lying above those of anatase for anatase—rutile heterophase junctions, illustrating a plausible charge separation and transfer processes across the phase junction.

Recently, Scanlon et al. reported theoretical and experimental studies on the band alignment between anatase and rutile of $TiO_2$ [30]. The authors found that the conduction band and valence band of the anatase phase were located below those of the rutile phase ($\Delta$ = 0.17 and 0.47 eV, respectively). The bandgap at the interface between anatase and rutile phases were experimentally confirmed by XPS analysis of heterophase $TiO_2$ nanoparticles and indicated a reduction in the bandgap to below 3 eV at the interface. Such staggered bandgap promotes the electron migration to anatase phase and hole transfer to the rutile phase.

On the other hand, for the heterophase photoanode, the photogenerated charge carriers should be separated and transported in the desired direction. Thus, the engineering sequence of the different phases in an appropriate band alignment of the heterophase junctions favoring charge separation/transfer at the interface is vital in a PEC system to minimize the electron-hole recombination and achieve efficient charge collection for high PEC activity. Apparently, for the reverse band alignment in type-II junctions, we expect the increase of the charge recombination, leading to low PEC performance due to inefficient charge separation/transfer. In an A-R heterophase junctions, the R-TNT/A-TNP photoanode with a right phase configuration corresponds to proper band alignment, while the A-TNT/R-TNP photoanode with a reverse phase configuration corresponds to improper band alignment.

To evaluate the PEC properties of single-phase and heterophase junction electrodes with TNT and TNP in various structural configurations and band alignments, linear weep voltammetry (LSV) measurements were performed in 1 M KOH (pH 13.8) aqueous electrolyte under 1 sun illumination (AM 1.5 G, 100 mW/cm$^2$). The potential is converted to the reversible hydrogen electrode (RHE) potential based on the Nernst equation: $\varphi_{RHE} = \varphi_{Ag/AgCl} + \varphi^{\circ}_{Ag/AgCl\ vs.\ SHE} + 0.059pH$. Here, $\varphi_{RHE}$ is the converted potential versus RHE, $\varphi_{Ag/AgCl}$ is the external potential measured against the Ag/AgCl reference electrode. $\varphi^{\circ}_{Ag/AgCl\ vs.\ SHE}$ is the potential of the Ag/AgCl reference electrode with respect to the standard hydrogen potential (SHE), and $\varphi^{\circ}_{Ag/AgCl\ vs.\ SHE}$ = 0.198 V at 25 °C.

Figure 5a shows *J-V* curves of R-TNT/A-TNP and A-TNP. The heterophase junctions with R-TNT/A-TNP structure achieves a photocurrent density of about 0.37 mA/cm$^2$ at 1.23 $V_{RHE}$, which is two times larger than that obtained for A-TNP photoanode (0.18 mA/cm$^2$ at 1.23 $V_{RHE}$). However, compared with R-TNP electrode (0.03 mA/cm$^2$ at 1.23 $V_{RHE}$), we observed one order of increase in photocurrent density of R-TNT/A-TNP (Figure S6a). Moreover, R-TNP/A-TNP heterophase junction with only TNP and the same phase configuration shows slightly less photocurrent density than that of R-TNT/A-TNP. Interestingly, A-TNT/R-TNP with a reverse band-alignment exhibits a photocurrent density of 0.01 mA/cm$^2$ at 1.23 $V_{RHE}$, which corresponds to the lowest value among the heterophase

junctions due to significant charge recombination, as mentioned above (Figure S6a). Note that all photoanodes have good transient photocurrent responses in chopped light cycles.

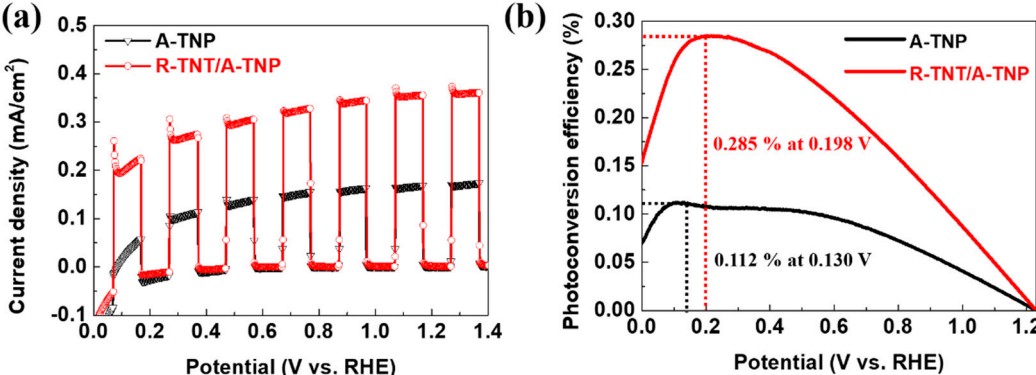

**Figure 5.** PEC performance of R-TNT/A-TNP and A-TNP: (**a**) *J-V* curves under chopped light illumination; (**b**) photoconversion efficiency as a function of applied potential.

When the light is off, the current reaches almost zero, while the photocurrent rapidly rises to correspondent values upon illumination, which is reproducible with almost identical photocurrent and dark current under light on/off cycles.

Regarding PEC water oxidation, the remarkable enhancement of photocurrent for R-TNT/A-TNP heterophase junction photoanode demonstrates the important of nanoporous structures as well as appropriate band alignment through phase arrangement for achieving efficient charge separation/transfer. In addition to phase configuration, the lowering bandgap in the heterophase junction as compared with those of single phases also contribute to the expanded light absorption and the enhanced photoresponse.

The PEC performance of the heterophase junction photoanode was further evaluated by quantitative calculation of the photoconversion efficiency. Figure 5b shows photoconversion efficiency as a function of applied bias potential ($V_{RHE}$). Applied bias photon-to-current conversion efficiency ($\eta_{APE}$) of the photoanodes was calculated from *J-V* curves, assuming 100% Faradaic efficiency through the following equation: $\eta_{APE}$ (%) = $J_{ph}(V_{rev} - V_{RHE})/P_{light}$, where $J_{ph}$ is the photocurrent density (mA/cm$^2$), $V_{rev}$ is the standard reversible potential, which is 1.23 V vs. RHE, $V_{RHE}$ is the applied bias potential versus reversible hydrogen electrode, and $P_{light}$ is the power density of the incident light irradiance (100 mW/cm$^2$). The R-TNT/A-TNP achieves $\eta_{APE}$ = 0.285% at 0.198 $V_{RHE}$, which is 2.5, 20, and 142 times those obtained for A-TNP ($\eta_{APE}$ = 0.112% at 0.130 $V_{RHE}$), R-TNP ($\eta_{APE}$ = 0.013% at 0.556 $V_{RHE}$), and A-TNT/R-TNT ($\eta_{APE}$ = 0.002% at 0.820 $V_{RHE}$), respectively (Figure 5b and Figure S6b). The drastic improvement in photoconversion efficiency achieved at a low bias for R-TNT/A-TNP can be attributed to the nanostructured heterophase junctions with the appropriate band alignment and the lowering bandgap.

Electrochemical impedance spectroscopy (EIS) provides a useful technique for studying the charge transfer and recombination processes at photoelectrode–electrolyte interfaces. EIS measurement was performed in a frequency range of 0.01–500 kHz at 1 M KOH electrolyte under 1 sun illumination. Figure 6a shows the comparison of Nyquist plots for R-TNT/A-TNP and A-TNP photoanodes, demonstrating the beginning of a large semicircle associated with resistive and capacitive processes. The relationship between *Z'* (the real part) and *Z"* (the imaginary part) of a semicircle with a radius of *R*/2 can be expressed as follows: $(Z' - R/2)^2 + Z''^2 = (R/2)^2$. The relative size of a circular arc radius corresponds to the charge transfer resistance and electron-hole separation efficiency [35,36]. We found that the diameter of the semicircle for R-TNT/A-TNP under illumination in the high-to-medium frequency region is much smaller than that for A-TNP, which indicates a recombination suppression via an improved charge transfer and more effective charge separation.

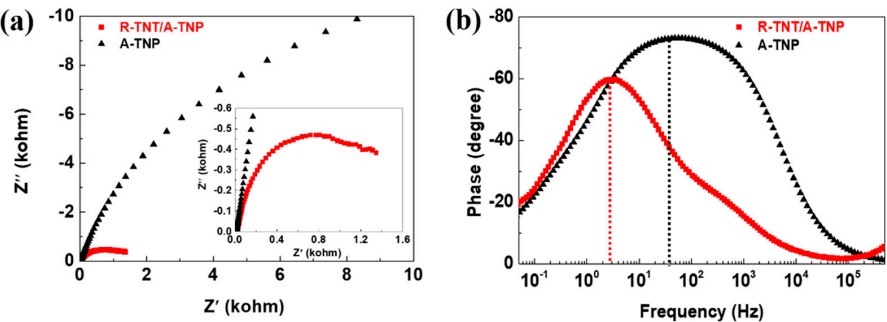

**Figure 6.** Electrochemical impedance spectroscopy of R-TNT/A-TNP and A-TNP: (**a**) Nyquist plots of photoanodes collected at 0.7 $V_{RHE}$ in 1M KOH (pH~14) electrolyte under 1 sun illumination. Inset is a zoomed-in spectra at high frequencies. (**b**) Corresponding Bode plots of photoanodes.

Furthermore, the Bode plots for R-TNT/A-TNP and A-TNP photoanodes under illumination indicate the presence of time constants at the low- and middle-frequency peaks, corresponding to the diffusion in the electrolyte and the electron transport-recombination process, respectively (Figure 6b). The electron lifetime ($\tau_e$) in the photoanodes can be obtained from the characteristic angular frequency ($\omega$) of the middle frequency ($f$) peak in the Bode phase plots by using the relation of $\tau_e = 1/\omega = 1/2\ \pi f$. The characteristic frequency for R-TNT/A-TNP and A-TNP were 3.37 and 56.60 Hz, which correspond to the electron lifetimes of 47.2 and 2.8 ms, respectively. It is noteworthy that the electron lifetime for R-TNT/A-TNP was 16.8 times than that for A-TNP. Therefore, it was obvious that R-TNT/A-TNP heterophase junctions showed low recombination rates as well as effective charge carrier separation, thus leading to high PEC activity.

In order to further understand the charge transport and recombination properties, the open-circuit potential transients was recorded with switching illumination on and off, which were accompanied by photovoltage relaxation from the illuminated quasi-equilibrium state to the dark equilibrium. In other words, as shown in Figure 7a, when switching the light on from a steady-state in the dark, the photoelectrode showed a negative shift of photovoltage due to photogenerated electron accumulation, thus resulting in rapid shifts of the Fermi level to positive potentials. Meantime, when the light is turned off, photovoltage decay was observed until the Fermi level returned to an original level because of the charge recombination. Obviously, open-circuit potential decay of R-TNT/A-TNP is slower than that of A-TNP, indicating moderate recombination kinetics in R-TNT/A-TNP. The corresponding response time clearly increased with photovoltage decay, thus indicating a long electron lifetime. The photovoltage decay rate by the following equation: $\tau_n = (K_B T/e)/d(Voc/dt)^{-1}$, where $\tau_n$ is the potential dependent electron lifetime, $K_B$ is the Boltzmann constant, $T$ is the Kelvin temperature, $e$ is the charge of a single electron, and $V_{oc}$ is the open-circuit potential at time $t$. As shown in Figure 7b, the R-TNT/A-TNP photoanode clearly presents a relatively longer electron lifetime than that of A-TNT. As a result, the heterophase junctions are beneficial for enhancing PEC activity of photoanodes.

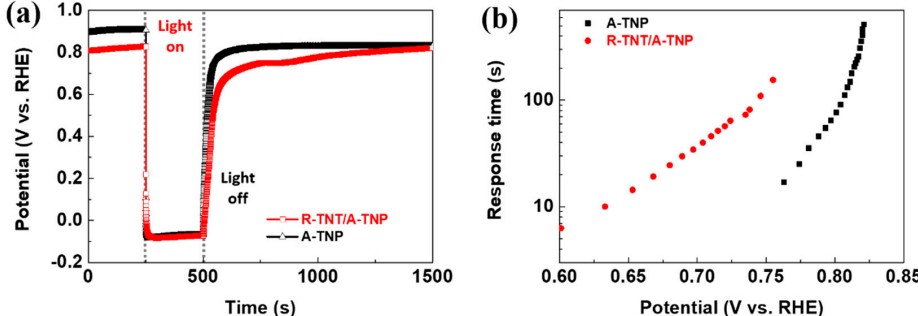

**Figure 7.** Charge transport properties of R-TNT/A-TNP and A-TNP: (**a**) Open-circuit potential vs. time profile of photoanodes; (**b**) electron response time vs. open-circuit potential plots of photoanodes.

## 4. Conclusions

In summary, we developed $TiO_2$-based heterophase junctions that have a larger specific surface area provided by nanostructures with high porosity and long aspect ratio; further, they exhibit an efficient charge transfer/separation offered by a favorable band alignment toward highly efficient PC water oxidation. By utilizing TNT and TNP nanostructures, the TNT/TNP heterophase junctions were fabricated with different phase sequences (A-TNT/R-TNP and R-TNT/A-TNP junctions). We investigated the PC properties of the heterophase junctions with respect to probe proper phase alignment in heterojunctions. The A-R heterophase junctions, considered to be a type-II band structure, present the lowering effective bandgap of ~2.9 eV. Benefitting from the efficient charge separation/transfer due to optimal band alignment and the enhanced surface catalytic activity due to 1D nanostructured TNT with hierarchical porous structure, the R-TNT/A-TNP junction exhibited superior PEC performance. The heterophase junction with R-TNT/A-TNP structure achieves a photocurrent density of about 0.37 mA/cm$^2$ at 1.23 $V_{RHE}$, which is two times and one order larger than those obtained for A-TNP (0.18 mA/cm$^2$) and for R-TNP (0.03 mA/cm$^2$), respectively. The R-TNT/A-TNP achieves $\eta_{APE} = 0.285\%$ at 0.198 $V_{RHE}$, which is 2.5, 20, and 142 times those obtained for A-TNP, R-TNP, and A-TNT/R-TNT, respectively. The remarkable improvement in photoconversion efficiency achieved at a low bias for R-TNT/A-TNP can be attributed to the nanostructured heterophase junctions with appropriate band alignment as well as the lowering bandgap. This work suggests that heterophase junction engineering can provide a promising approach in energy-conversion technology.

**Supplementary Materials:** The following are available online at http://www.mdpi.com/2079-6412/10/6/557/s1, Figure S1: The phase transformation of $TiO_2$ nanotubes with annealing temperatures by using rapid thermal treatments, Figure S2: The full XPS survey spectra of A-TNT (upper) and R-TNT (lower), respectively, Figure S3: XPS spectra of (a) O1s and (b) Ti2p for A-TNT and R-TNT, Figure S4: XPS sputter-depth profiling spectra of O1s and Ti2p peaks for heterophase junctions: (a) R-TNT/A-TNP and (b) A-TNT/A-TNP, Figure S5: The optical bandgap estimation of R-TNP and A-TNT/R-TNP through Tauc plots, Figure S6: The PEC performance of R-TNP/A-TNP, A-TNT/R-TNP, and A-TNP: (a) J-V curves under chopped light illumination and (b) the photoconversion efficiency as a functions of applied potential, Table S1: Comparison studies for various $TiO_2$ photoanodes and their PEC performances, Table S2: Summary of structural parameters for two different heterophase junctions obtained from XRD patterns.

**Author Contributions:** Conceptualization, T.C.; methodology, S.L. and A.Y.C.; validation, T.C.; formal analysis, S.L. and A.Y.C.; investigation, S.L., A.Y.C., Y.S.R., and J.-Y.P.; writing—original draft preparation, T.C., S.L., and A.Y.C.; writing—review and editing, T.C., S.L., and A.Y.C.; visualization, S.L., A.Y.C., Y.S.R., and J.-Y.P.; supervision, T.C. All authors have read and agreed to the published version of the manuscript.

**Funding:** This research was supported by Next Generation Engineering Researcher Program of National Research Foundation of Korea (NRF) funded by the Ministry of Science, ICT (NRF-2019H1D8A2106002) and also supported by the Korea Institute of Energy Technology Evaluation and Planning (KETEP) and the Ministry of Trade, Industry & Energy (MOTIE) of the Republic of Korea (No. 20184030202260).

**Conflicts of Interest:** The authors declare no conflict of interest.

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
