# Peer review of "Synergistic Design of Anatase–Rutile TiO2 Nanostructured Heterophase Junctions toward Efficient Photoelectrochemical Water Oxidation"

_coatings, doi:10.3390/coatings10060557_

Round 1
Reviewer 1 Report
In this article, authors explained photoelectrochemical (PEC) performance for solar-water splitting by using TiO2 heterophase junctions. the manuscript is well organized and explained indetailed.
So I am happy to accept this manuscript.
Author Response
Response to Reviewer 1 Comments
In this article, authors explained photoelectrochemical (PEC) performance for solar-water splitting by using TiO2 heterophase junctions. the manuscript is well organized and explained in detailed.
So I am happy to accept this manuscript.
Response: We highly appreciate the reviewer’s positive response on our manuscript.
Reviewer 2 Report
This manuscript needs following modifications,
- There are other reports on this topic and authors should provide a table comparing the performance of this study with other reports and discuss about the novelty of this study. One example: https://doi.org/10.1021/acssuschemeng.8b02130
- Clear SEM and TEM images must be provided.
- XPS study should be performed to be certain about the chemical composition of the material.
Reviewer 3 Report
- General : Please check the grammatical errors, wherever required. English needs to be refined/polished.
- There are spelling mistakes here and there. Please carefully edit.
- General: Authors are requested to unify the font usage throughout the manuscript.
- General: Symbols are in a different font.
- General: Take care of equations
- Abstract: change the premise to promise. Few sentences need clarity. Please revisit Abstract once again.
- Introduction: I suggest the authors to cite relevant articles pertaining to mixed-phase TiO2 (Nanomaterials 10 (3), 456, 2020, Catalysts 9 (2), 170, 2019, Materials Chemistry and Physics 154, 125-136, 2015).
- Change consisting with to “consisting of”
- 2(a): Include discussions pertaining to ID/IG ratio
- 2(b): XRD graph is not very clear.
- Fix this sentence: “We thus take use of Tauc plots”.
- Identify other similar occurrences and correct the same.
- If possible, provide survey and core-level XPS data for R-TNT/A-TNP and A-TNP.
